# Association between Time to Local Tumor Control and Treatment Outcomes Following Repeated Loco-Regional Treatment Session in Patients with Hepatocellular Carcinoma: A Retrospective, Single-Center Study

**DOI:** 10.3390/life11101062

**Published:** 2021-10-09

**Authors:** Krzysztof Bartnik, Wacław Hołówko, Olgierd Rowiński

**Affiliations:** 1Doctoral School, Medical University of Warsaw, 02-091 Warsaw, Poland; krzysztof.bartnik@wum.edu.pl; 2Second Department of Radiology, Medical University of Warsaw, 02-091 Warsaw, Poland; olgierd.rowinski@wum.edu.pl; 3Department of General, Transplant and Liver Surgery, Medical University of Warsaw, 02-091 Warsaw, Poland

**Keywords:** transarterial chemoembolization, ablation, hepatocellular carcinoma, time, local tumor control, complete remission

## Abstract

Background: Whether the number of loco-regional treatment sessions and the time required to obtain local tumor control (LTC) affects the prognosis of patients with hepatocellular carcinoma (HCC) remains controversial. This study aimed to determine whether a longer time to LTC is a significant and independent predictor of poor treatment outcomes. Methods: In this retrospective study, we analyzed data of 139 treatment-naive patients with HCC who were not eligible for a treatment other than transarterial chemoembolization (TACE) at baseline. The outcome analyses were performed using the Cox proportional hazard model and Kaplan–Meier method, while the overall survival (OS) and progression free survival (PFS) were the primary study endpoints. Results: Overall, LTC was achieved in 82 (59%) of patients, including 67 (81%) patients who achieved LTC following TACE sessions alone and 15 (19%) subjects required additional ablation session. The median OS did not differ significantly between groups that needed 2, 3, or >3 locoregional treatment sessions to achieve LTC (*p* = 0.37). Longer time to LTC (in weeks) was significantly associated with shorter OS in univariate analysis (*p* = 0.04), but not in an adjusted model (*p* = 0.14). Both univariate and adjusted analyses showed that longer time to reach LTC was significantly associated with shorter PFS (adjusted HR = 1.04, 95% CI 1.001–1.09, *p* = 0.048). Conclusions: These findings show that the longer time to LTC is not an independent predictor of OS, but suggest that PFS may be significantly shorter in patients with longer time to LTC.

## 1. Introduction

Therapy for hepatocellular carcinoma (HCC) consists of various treatment modalities, including liver transplantation, resection, systemic treatment, and locoregional therapies [1,2]. Despite the tremendous progress that has been made in the treatment of HCC, the prognosis remains very poor as curative treatment options are available only for a small group of patients with early-stage HCC. Therefore, transarterial chemoembolization (TACE) is currently recommended as the first-line treatment modality for many patients in whom curative treatment is not feasible [3]. Standard TACE therapy consists of repeated treatment sessions with follow-up imaging by computed tomography (CT) or magnetic resonance imaging (MRI) [4,5]. Multiple TACE sessions alone or combined with other locoregional treatments are necessary for most patients to achieve local tumor control (LTC) [6,7]. Lack of viable tumor tissue is the desired treatment outcome, but this is unfortunately not achieved in all patients [7,8].

Previous studies that examined the association between early LTC and long-term clinical outcomes have shown conflicting results [7,9,10,11,12]. To date, there has been little agreement on whether a lack of early tumor control should lead to a change in treatment strategy (e.g., discontinuation of locoregional treatment with subsequent systemic therapy) [13,14]. In addition, experience with neoplasms other than HCC shows that the time interval between initiation of treatment and tumor control significantly impacts patient outcomes, indicating that more research is needed to optimize locoregional therapy in HCC patients [15,16].

This study aims to determine whether a longer time to LTC is a significant and independent predictor of poor treatment outcomes. For this purpose, we analyzed data for treatment-naive patients with HCC who achieved LTC after repeated locoregional sessions.

## 2. Materials and Methods

In this retrospective study, we analyzed consecutive treatment-naive patients with HCC confirmed by the imaging criteria by the American Association for the Study of Liver Diseases who underwent initially at least two conventional TACE sessions between 2016 and 2018. The inclusion criteria were as follows: (1) treatment-naïve patients with at least one HCC lesion, (2) patients were not eligible for other than TACE treatment at baseline, (3) patients had Barcelona Clinic Liver Cancer (BCLC) stage A/B disease, (4) patients had Child–Pugh score (CPS) of A or B, (5) patients had an available dynamic contrast-enhanced CT or MRI liver examinations. The exclusion criteria were: (1) patients that did not complete at least two TACE sessions, (2) other than TACE or ablation HCC-specific invasive treatment (e.g., resection, liver transplantation), (3) other malignant neoplasm or uncontrolled disease during follow-up, and (4) incomplete clinical data. The final study population and a summary of the enrollment criteria are shown in Figure 1. The study was approved by the local Institutional Ethical Committee of Human Experimentation and was compliant with the current version of the Helsinki Declaration.

All patients underwent at least two conventional TACE procedures. During the procedure, 20–40 mL of a 1:1 mixture of lipiodol and doxorubicin was selectively injected into the tumor feeding artery until flow stasis was achieved, with subsequent embolization with gelatin sponge particles (Spongostan absorbable hemostatic gelatin sponge, Ethicon Inc.). 

A standard treatment cycle consisted of at least two sessions of TACE at 4–6-week intervals with subsequent imaging (30–90 days after second (or more if indicated) TACE). If a viable tumor was evident, an additional TACE or ablation procedure was performed according to “treatment stage migration” strategy [17,18]. Patients not eligible for further locoregional treatment (TACE or ablation) were referred for sorafenib treatment or palliative care (if systemic treatment was not feasible).

Overall, 29 patients needed at least 1 percutaneous radiofrequency ablation (RFA) or microwave ablation (MVA) under general anesthesia following initial TACE sessions. The type of ablation needle was selected depending on the target size of the lesion, and then ablation was performed for 4–10 min. Subsequent ultrasound or CT examination was performed immediately after the ablation to exclude active bleeding or damage to adjacent organ.

Follow-up examinations were performed by CT or MRI according to the LI-RADS 2018 technical recommendations, and treatment responses were assessed using the LI-RADS treatment response (LR-TR) algorithm [19]. LTC was defined as a complete remission with nonviable tumor response (using LR-TR criteria) of all treated HCC lesions (a per session manner). If the patient had more than one treated HCC lesions with different treatment responses, the final response category was reported in aggregate, selecting the least favorable response.

Baseline demographic data, laboratory tests, liver function (CPS and albumin-bilirubin grade (ALBI)), and tumor (BCLC) stage were analyzed. The date of the first TACE session was set as an index day. The date of LTC was specified as the date of the last TACE (or ablation) procedure before response evaluation. If a LTC was achieved, patients were followed up by repeated imaging and measurements of serum α-fetoprotein concentration until HCC recurrence was confirmed. The date of death or last clinical follow-up was defined as the end of the follow-up period. Overall survival (OS) was the primary study endpoint. In patients who achieved LTC, PFS was defined as the interval between reaching LTC and the date of reported progression (including detection of new intrahepatic HCC lesions).

Kruskal–Wallis and Fisher’s exact tests were used to compare differences between the studied subgroups. Kaplan–Meier method and Log-rank tests with Šidák multiple comparison correction were used to compare survival curves between study subgroups. Univariate and multivariate survival analyses were performed using Cox proportional hazards models. The hazard radios (HRs) in Cox regression models were adjusted for predictors with *p*  ≤  0.2 derived from univariate regression analyses. SAS software (Statistical Analysis System version 9.4, SAS Institute Inc., Cary, NC, USA) was used for all statistical analyses and artwork.

## 3. Results

### 3.1. Patient Characteristics

Final analyses included 139 patients, 82 (59%) of whom achieved LTC. Overall, 67 (81%) patients achieved LTC following TACE sessions alone, while 15 (19%) patients required additional ablation treatment following initial TACE sessions. Subjects who achieved LTC had less advanced BCLC stage, had with fewer treated tumor lesions of a smaller dimension, had a lower percentage of patients with baseline alpha-fetoprotein (AFP) concentration below 200 ng/mL and lower baseline AST level (Table 1). Patients that required >3 locoregional sessions to achieve LTC had significantly more advanced BCLC stage than the rest of patients. Overall, 9 patients had infiltrative HCC type, but none of them achieved LTC. The remaining variables did not differ between the analyzed groups. Clinical characteristics of patients that achieved LTC is shown in Table 2. The median follow-up was 29 months while the median OS of the entire cohort was 33 months. The median OS was significantly longer in patients who achieved LTC than in those who did not (51 vs. 16 months, *p* < 0.001) (Figure 2). The median OS did not differ significantly between groups with different number of locoregional treatment sessions in the entire cohort (*p* = 0.93).

### 3.2. Treatment Outcomes in Patients with No LTC

In a subgroup of 57 patients with no LTC, locoregional treatment was discontinued after 2, 3, and >3 sessions in 25 (49%), 18 (27%), and 14 (24%) patients, respectively. In those patients locoregional treatments was discontinued due to: untreatable tumor progression in 37 patients (65%) with subsequent palliative or sorafenib treatment (19 and 18 subjects, respectively); portal vein thrombosis in 10 patients (17%); decompensation of liver function in 6 patients (11%); other reasons (7%). The median OS was not significantly associated with the number of TACE sessions in patients with no LTC (*p* = 0.17).

### 3.3. Treatment Outcomes in Patients LTC

Among those who achieved LTC, 42 achieved complete remission after two courses of TACE, whereas 26 required three sessions and 14 required more than three sessions of locoregional treatment. LTC was achieved by 14 (50%) of the 28 patients who received more than three sessions of locoregional therapy. The median OS was 53 months for patients with LTC requiring two sessions of TACE, 60 months for those requiring three sessions, and 39 months for those requiring more than three sessions (*p* = 0.37; Figure 3). In a multivariate model fitted for BCLC, adjusted HRs for OS were 0.81 (95% CI 0.36–1.81, *p* = 0.32) for three sessions and 1.44 (95% CI 0.6–3.45, *p* = 0.27) for more than three versus two sessions. 

Overall, the median PFS was 21 month (95% CI 16–26), while 48 patients (58%) had confirmed progression and 34 individuals (42%) were censored during follow-up period. The KM survival analysis showed that the median PFS did not differ between the LTC groups (21 months for patients requiring two sessions of TACE, 24 months for those requiring three sessions, and 30 months for patients requiring more than three sessions (*p* = 0.69; Figure 4).

### 3.4. Time to LTC as a Prognostic Factor

A median of 10 weeks (range 5– to 52) between 1st treatment session and LTC was observed. Hazard ratios for the prediction of OS and PFS for prognostic factors generated from the univariate analysis are shown in Table 3. Univariate analysis showed that the BCLC B stage and longer time (in weeks) between treatment initiation and achieving LTC were a significant risk factor for early death. However, after accounting for BCLC stage an adjusted model showed no significant association between longer time to LTC and OS (HR = 1.02, 95% CI 0.99–1.05, *p* = 0.14). 

The univariate Cox regression model exploring prognostic factors for PFS is shown in Table 3. Longer time to LTC (in weeks), as well as BCLC stage B, were significantly associated with shorter PFS. After accounting for BCLC and CPS score, multivariate analysis revealed a significant independence of the longer time to LTC as predictor of shorter PFS (HR = 1.04, 95% CI 1.001–1.09, *p* = 0.048).

## 4. Discussion

Liver transplantation and liver resection are the most efficient therapeutic alternatives for both HCC recurrence and patients survival, thus should be offer to all eligible subjects [20]. However, only patients diagnosed at an early HCC stage would benefit from treatments with curative intent. The present study assessed the outcomes of patients with HCC ineligible for therapies other than TACE or ablation using a treatment stage migration strategy [21].

It has become increasingly clear that post-treatment data may provide important prognostic information that can significantly refine risk stratification in patients with HCC treated with TACE [20,21]. This potentially entails an assessment of the dynamic response of HCC to TACE or other locoregional treatments, which may not be fully captured by baseline analysis [6,19]. In the present study, time to LTC and the number of treatment sessions required to reach LTC were analyzed as predictors of OS and PFS in patients with HCC that were not eligible for other than TACE treatment at baseline. The most remarkable result to emerge from our data is that longer time to LTC (in weeks) was significantly associated with shorter PFS. Interestingly, our results suggest no association between longer LTC and OS after accounting for disease stage (BCLC) arguably the most important prognostic factor for patients with HCC undergoing locoregional therapies [22,23,24]. Moreover, these findings suggest that early tumor control, as estimated by the number of sessions of treatment sessions or time required to achieve a nonviable response, was not independently associated with OS. Similarly, results presented by Park et al. show no significant difference in OS comparing initial complete response patients with subsequent complete response group [25]. Thus, we hypothesize that the association of poorer patient outcomes with a lack of complete response after first locoregional session may be due to unfavorable baseline risk factors. It is worth to underline that the study group included only LR-5 tumors, although Centonze et al. suggest that including tumors LR-3 and LR-4 in OS prediction does not significantly decrease its precision [26]. Moreover, presented results extend recent suggestions by Wang et al. that the best objective response correlates better with treatment outcomes in patients with more advanced disease, as the concordance between initial and overall best response is weak in those subjects [7]. It is worth noting that only patients with at least two TACE sessions were included in our analysis. In our center, standard TACE therapy consists of at least 2 treatment sessions followed by a response evaluation (using CT or MRI). We do not routinely examine patients with imaging studies after the first treatment session, but refer them to a second TACE session approximately 4 weeks apart, where we analyze the feasibility of embolization with angiography and cone beam computed tomography. Patients with complications (e.g., deterioration in liver function) or clinical progression after the first TACE were disqualified from further TACE and were not included in this analysis. Exclusion of patients that did not complete at least two TACE sessions may potentially affect the outcome analysis of a no-LTC subgroup in an intention to treat perspective and potentially create a selections bias.

Previous findings on the relevance of an early response have been inconsistent and contradictory. Several studies demonstrated that an initial favorable response is a robust predictor of survival following TACE in patients with HCC. For example, Kim et al. showed that the initial, as well as the best overall response, are reliable OS predictors while achievement of early tumor control was the most effective predictor for favorable treatment outcomes [9]. Another analysis by Kim et al. showed non-responder by the mRECIST criteria one month after initial TACE was an independent and significant prognostic factor for OS [11]. However, some authors have questioned this by finding no association between initial treatment response and long-term clinical outcomes after adjusting for confounding factors [27,28]. Similarly, no association between number of procedures nor time to achieve LTC and OS was found in the present study. This is in line with our recent study in which an adjusted analysis showed no significant prognostic value of the initial LR-TR response despite improved OS in the nonviable subgroup in the unadjusted model [29].

Overall, a significant proportion of patients did not respond well to the initial TACE sessions (only 30% of patients achieved local tumor control following two TACE sessions alone), keeping with the study’s results by Georgiades et al. [8]. Of note, many patients who did not respond to the initial TACE session responded favorably after the subsequent sessions, with 59% of patients achieving an overall LTC. Since a large proportion of patients will achieve LTC after more than three sessions of locoregional treatment sessions and response following the third and subsequent sessions cannot be predicted, our results suggest that treatment outcomes may not be significantly worse in patients who do not achieve early LTC. Importantly, as there is a strong association between LTC and OS, further improvement in treatment outcomes can be achieved by increasing the proportion of patients achieving complete radiological remission. There is a need for study to asses if closer clinical follow-up of patients with longer time to LTC may improve their treatment outcomes.

The median OS of patients with BCLC B stage in our cohort was 22 month after initiation of treatment, which is in line with previous reports, including the study by Biolato et al. in which the OS of the entire cohort of patients with stage B BCLC was 23 months [30]. A study by Burrel et al. showed that the median OS can be extended to 48.6 months with careful patient selection [31]. The shorter median OS observed in our cohort (33 month) could be explained at least in part by the fact that we only included patients who were not eligible for OLTx or resection. These results are in line with a recent multicenter analysis in which the median OS of the entire TACE group ranged from 13.7 to 33.8 months [32]. A recent randomized controlled trial by Kuda et al. showed that the median time to untreatable tumor progression in patients treated with TACE alone was 20.6 months, which correlates with a median PFS of 21 months in the LTC cohort in the present study. Our results suggest that longer time to LTC (in weeks) may be significantly associated with a shorter PFS. However, given that our findings are based on a limited number of 82 patients the preliminary results from such analyses should be interpreted with caution. A prospective randomized trial will be necessary to definitively address the association between early LTC and long-term clinical outcomes in HCC patients treated with multiple sessions of locoregional therapy.

Our work clearly has some limitations. First, it was not possible to analyze the cause of death for all patients with LTC in our cohort. Overall, more than half of LTC subjects had confirmed progression, while the rest were lost during follow-up period. Second, only treatment-naive patients without extrahepatic disease and baseline portal vein thrombosis were included in the analysis. Moreover, patients who underwent ablation before TACE, resection or liver transplantation were excluded from the analysis. Such exclusion criteria increase the homogeneity of the study group; however, this may limit the generalizability of our findings, as HCC treatment is usually multimodal, and the outcomes of patients treated are heterogeneous. Importantly, future efforts to improve patient outcomes should focus on changing modifiable risk factors and identifying patients who may benefit from treatment stage migration strategy [33,34]. Of note, we did not analyze the incidence of TACE complications. Thus, we do not know whether or not more TACE sessions will significantly increase the incidence of such adverse events. However, it is unlikely that it would have significantly influenced the results of this study as median OS was not associated with the number of TACE sessions in any of the analyzed groups: with or without LTC. Nevertheless, it would be interesting to perform an analysis adjusted for complications after repeated TACE sessions.

## 5. Conclusions

The results of this study indicate that the time between treatment initiation and achieving LTC adds no prognostic information for the OS of patients with HCC who are not eligible for treatment other than TACE at baseline. However, the evidence from this study suggests that PFS may be significantly shorter in patients with longer time to LTC and there is a need for further analysis to clarify whether these patients may benefit from closer clinical follow-up.

## Figures and Tables

**Figure 1 life-11-01062-f001:**
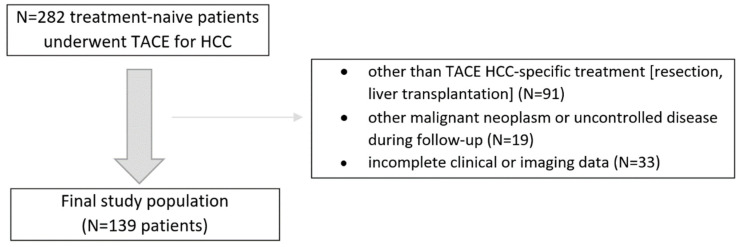
Flowchart of the study population enrollment.

**Figure 2 life-11-01062-f002:**
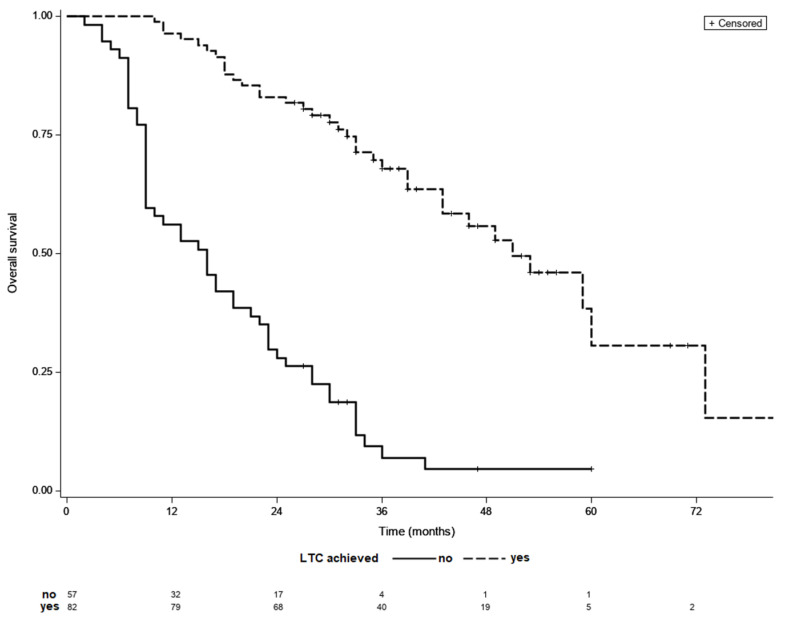
Kaplan–Meier survival curves of patients with and without LCT; median OS 51 vs. 16 months, *p* < 0.001 (log-rank test).

**Figure 3 life-11-01062-f003:**
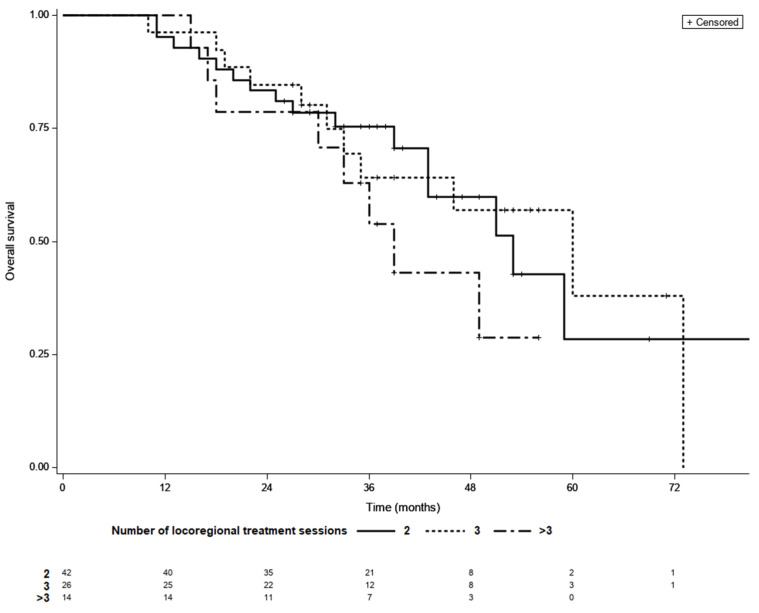
Kaplan–Meier survival curves of patients with different number of locoregional treatment sessions needed to achieve LTC; 2 vs. 3—*p*  =  0.98, 2 vs. >3—*p*  =  0.79, 3 vs. >3—*p*  =  0.95 (log-rank test, Šidák correction).

**Figure 4 life-11-01062-f004:**
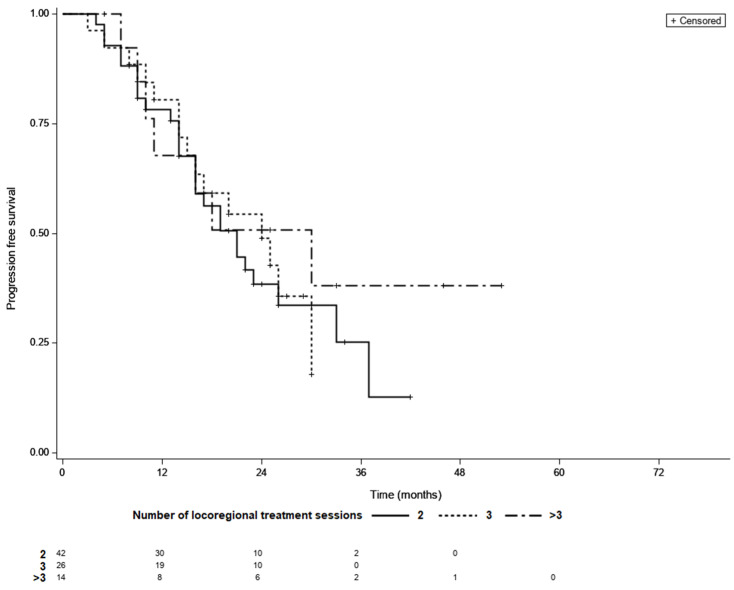
Progression free survival curves of patients with different number of locoregional treatment sessions needed to achieve LTC; LTC; 2 vs. 3—*p*  =  0.99, 2 vs. >3—*p*  =  0.66, 3 vs. >3—*p*  =  0.62 (log-rank test, Šidák correction).

**Table 1 life-11-01062-t001:** Demographic and clinical characteristics of patients in subgroups defined by status of LTC.

Variable	No LTC	LTC	*p* Value
N of patients	57	82	-
Age (years)			
<60	17 (30%)	21 (26%)	0.56
>60	39 (70%)	61 (74%)	
Gender			
Male	46 (81%)	58 (71%)	0.23
Female	11 (19%)	24 (29%)	
BCLC stage			
A	16 (28%)	51 (62%)	<0.001
B	41 (72%)	31 (38%)	
Child Turcotte Pugh Class			
A	48 (84%)	75 (91%)	0.27
B	9 (16%)	7 (9%)	
Serum AFP			
<200 ng/mL	32 (56%)	64 (78%)	0.01
≥200 ng/mL	25 (44%)	18 (22%)	
ALBI			
1	27 (47%)	51 (62%)	0.16
2	26 (46%)	29 (35%)	
3	4 (7%)	2 (2%)	
Albumin, g/L	3.95 (2.5–5.0)	4.1 (2.0–5.2)	0.17
Creatinine	0.87 (0.56–1.56)	0.91 (0.54–1.97)	0.65
Total bilirubin (umol/L)	0.98 (0.3–5.6)	1.05 (0.21–4.04)	0.14
INR	1.15 (0.91-1.66)	1.15 (0.92–2.45)	0.82
ALT, IU/L	47 (12–348)	49 (19–303)	0.69
AST, IU/L	65 (18–408)	50 (20–423)	0.02
N of treated HCC lesions			
1	22 (39%)	55 (67%)	0.003
2	17 (30%)	19 (23%)	
≥3	18 (31%)	4 (10%)	
Tumor size			
<30 mm	9 (16%)	32 (39%)	0.002
30–50 mm	20 (35%)	31 (38%)	
>50 mm	28 (49%)	19 (23%)	
Fulfilled Milan criteria			
Yes	6 (11%)	21 (26%)	0.03
No	51 (89%)	61 (74%)	

**Table 2 life-11-01062-t002:** Demographic and clinical characteristics of patients in subgroups defined by number of sessions needed to achieve LTC.

Variable	2 Sessions	3 Sessions	>3 Sessions	*p* Value
N of patients	42	26	14	-
Age (years)				
<60	13 (31%)	5 (19%)	3 (21%)	0.59
>60	29 (69%)	21 (81%)	11 (79%)	
Gender				
Male	27 (64%)	22 (85%)	9 (64%)	0.16
Female	15 (36%)	4 (15%)	5 (36%)	
Child Turcotte Pugh Class				
A	40 (95%)	24 (92%)	11 (79%)	0.16
B	2 (5%)	2 (8%)	3 (23%)	
BCLC Stage				
A	30 (71%)	16 (62%)	5 (36%)	0.06
B	12 (29%)	10 (38%)	9 (64%)	
Serum AFP				
<200 ng/mL	32 (76%)	21 (81%)	11 (79%)	0.94
≥200 ng/mL	10 (24%)	5 (19%)	3 (21%)	
ALBI				
1	27 (64%)	17 (65%)	7 (50%)	0.73
2	14(33%)	8 (31%)	7 (50%)	
3	1 (3%)	1 (4%)	0 (0%)	
Albumin, g/L	4.15 (3.0–5.2)	4.3 (2.0–5.2)	3.9 (2.8–4.6)	0.16
Creatinine	0.89 (0.54–1.97)	0.92 (0.65–1.35)	0.89 (0.59–1.42)	0.81
Total bilirubin (umol/L)	0.85 (0.24–4.04)	0.98 (0.43–2.0)	1.01 (0.42–3.0)	0.76
INR	1.15 (0.92–2.45)	1.16 (1.01–1.69)	1.17 (0.97–1.43)	0.97
ALT, IU/L	49 (19–200)	49 (20–303)	55 (23–264)	0.6
AST, IU/L	42 (20–189)	50 (22–243)	72 (22–193)	0.14
N of treated HCC lesions				
1	32 (76%)	16 (62%)	7 (50%)	0.31
2	7 (17%)	8 (31%)	4 (29%)	
≥3	3 (7%)	2 (7%)	3 (21%)	
Tumor size				
<30 mm	19 (45%)	9 (35%)	4 (29%)	0.07
30–50 mm	18 (43%)	10 (38%)	3 (21%)	
>50 mm	5 (12%)	7 (27%)	7 (50%)	

**Table 3 life-11-01062-t003:** Hazard ratios for the prediction of OS and PFS for prognostic factors generated from the univariate analysis.

Variable	Hazard Ratio	95% Confidence Interval	*p* Value
**Overall survival**			
LTC × 2 sessions	1.00		reference
LTC × 3 sessions	0.96	0.76–2.36	0.41
LTC × >3 sessions	1.74	1.00–1.01	0.16
Time to LTC (weeks)	1.03	1.01–1.6	0.04
BCLC A	1.00		reference
BCLC B	2.06	1.06–4.02	0.03
CPS A	1.00		reference
CPS B	1.87	0.64–5.38	0.25
ALBI grade 1	1.00		reference
ALBI grade 2	1.73	0.87–3.44	0.93
ALBI grade 3	3.25	0.42–24.84	0.38
**Progression free survival**			
LTC × 2 sessions	1.00		reference
LTC × 3 sessions	0.92	0.49–1.75	0.76
LTC × >3 sessions	0.69	0.30–1.61	0.43
Time to LTC (weeks)	1.05	1.02–1.09	0.004
BCLC A	1.00		reference
BCLC B	1.86	1.04–3.35	0.04
CPS A	1.00		reference
CPS B	2.02	0.79–5.16	0.14
ALBI grade 1	1.00		reference
ALBI grade 2	1.48	0.82–2.67	0.68
ALBI grade 3	3.16	0.74–13.41	0.19

## Data Availability

The datasets generated or analyzed during the current study are available from the corresponding author on reasonable request.

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
