# Peer review of "Association between Time to Local Tumor Control and Treatment Outcomes Following Repeated Loco-Regional Treatment Session in Patients with Hepatocellular Carcinoma: A Retrospective, Single-Center Study"

_life, 2021, doi:10.3390/life11101062_

Round 1

Reviewer 1 Report

General comments

In this study, the authors retrospectively evaluated the clinical implications of time to local tumor control (LTC) and treatment outcomes during repeated TACE for HCC. They included 139 treatment-naive patients with HCC who were not eligible for a 15 treatment other than TACE at baseline.

They found that longer time to LTC was significantly associated with shorter 22 OS in univariate analysis (p=0.04), but not in an adjusted model (p=0.14). Both univariate and ad-23 justed analyzes showed that longer time to reach LTC was significantly associated with shorter PFS 24 (adjusted HR=1.04, 95% CI 1.001-1.09, p=0.048).

Based on these results, the authors concluded that the longer time to LTC is not an independent predictor of OS, but suggest that PFS may be significantly shorter 26 in patients with longer time to LTC.

It is well written, the data and conclusions are well presented. The conclusions are useful to our daily practice and will likely impact patient outcomes in a positive way.

Specific comments

Materials & Methods

How many cases of portal vein tumor thrombosis (PVTT) or infiltrative tumor type are included (not excluded) in this study? PVTT and infiltrative tumor type are well-known poor prognostic factor for patients with HCC receiving repeated TACE. In these cases, the complete local tumor control is very difficult and required multisession TACE. Further subgroups analysis including PVTT / infiltrative tumor type is recommended.

Tables & figures

Please provide numbers at risks for all survival curve.

Discussion

The clinical implications of time to local tumor control (LCT) is not new and has been evaluated in a number of previous studies as the authors state in the discussion. One of the recent study by Park et al. (J Vasc Interv Radiol. 2020 Dec;31(12):1998-2006.e1.), describe both initial and best response during repeated TACE were strongly associated with clinical outcomes in patients with intermediate-stage HCC. Consider inclusion of these findings into the discussion.

Author Response

How many cases of portal vein tumor thrombosis (PVTT) or infiltrative tumor type are included (not excluded) in this study? PVTT and infiltrative tumor type are well-known poor prognostic factor for patients with HCC receiving repeated TACE. In these cases, the complete local tumor control is very difficult and required multisession TACE. Further subgroups analysis including PVTT / infiltrative tumor type is recommended.

Thank you for this comment. Overall, 9 patients had infiltrative tumor type, but none of them achieved LTC during the study period – we added this information in “patients characteristics” section. Moreover, none of patients had PVTT at baseline (BCLC C stage), as only patients with BCLC A/B stage were included. We agree that this issue is particularly relevant to the study results and we
added PVTT as a separate criterion to the exclusion criteria section. We did not observed PVTT in a LTC subgroup. Ten patients developed portal vein thrombosis before achieving LTC and were assigned to no-LTC subgroup – 50% of them had tumor-in-vein appearance. This fact probably contributes to the survival difference between LTC and non-LTC.

Please provide numbers at risks for all survival curve.

We added at risk numbers to all survival graphs.

The clinical implications of time to local tumor control (LCT) is not new and has been evaluated in a number of previous studies as the authors state in the discussion. One of the recent study by Park et al. (J Vasc Interv Radiol. 2020 Dec; 31(12):1998-2006.e1.), describe both initial and best response during repeated TACE were strongly associated with clinical outcomes in patients with intermediate stage HCC. Consider inclusion of these findings into the discussion.

Cited as advised.

Reviewer 2 Report

In their paper entitled “Association between time to local tumor control and treatment outcomes following repeated loco-regional treatment session in patients with hepatocellular carcinoma: a retrospective, single-center study” Bartnik and colleagues report their experience of repeated LRT for HCC management

This is an interesting study that report the results of the iterative approaches for HCC treatment: such attitute is progressively employed worldwide, so the study results are very important

I have fee suggestions to potentially improve the paper

1) the Authrs should better discuss the treatment allocation, as surgery and transplantstio remains the best treatments that (according to treatment migration) should be offered to hcc patients

2) the Authors should provide the concordance with the Milan criteria

3) was a Milan-out HCC considered a contraindication to LT? If so, what about those patients (how many?) achieving a LTC and downstaged to a Milan in HCC - none of them eligilble for LT?

4) the Authos should bette emphasize the exclusion criterikn reported in lines 209/211, as the exclusion of patients that did not completed the TACE cylcle strongly affects the results in an intention to treat perspective - such limitation should also be highlighted in the discussion

5) Reference 1 and reference 23 are the same, please correct

6) consider citing the following paper that focused on the outcomes of LT for HCC according to Metroticket score and Li-RADS 10.1111/tri.13983

Best regards

Author Response

1) the Authors should better discuss the treatment allocation, as surgery and transplantation remains the best treatments that (according to treatment migration) should be offered to hcc patients

Thank you for this comment. We added the following sentences into the discussion section: “Liver transplantation and liver resection are the most efficient therapeutic alternatives for both HCC recurrence and patients survival, thus should be offer to all eligible subjects. However, only patients diagnosed at an early HCC stage would benefit from treatments with curative intent. The present study assessed the outcomes of patients with HCC ineligible for therapies other than TACE or ablation using a treatment stage migration strategy.”

2) the Authors should provide the concordance with the Milan criteria

Table 1 has been updated with this data

3) was a Milan-out HCC considered a contraindication to LT? If so, what about those patients (how many?) achieving a LTC and downstaged to a Milan in HCC - none of them eligible for LT?

Downstaging to Milan Criteria with locoregional therapies is a strategy that can be applied to patients initially not eligible for liver transplantation. Unfortunately, the analysis of number of patients that were downstaged and subsequently transplanted or resected was beyond the scope of the current study. We did not collect the data of LT patients.

4) the Authors should better emphasize the exclusion criterias reported in lines 209/211, as the exclusion of patients that did not completed the TACE cylcle strongly affects the results in an intention to treat perspective - such limitation should also be highlighted in the discussion

We added additional exclusion criteria as follows: “1) patients that did not complete at least two TACE sessions,”. Moreover, We added the following sentence into the discussion with aim to highlight the importance of such exclusion criterium: “Exclusion of patients that did not complete at least two TACE sessions may potentially affect the outcome analysis of a no-LTC subgroup in an intention to treat perspective and potentially create a selection bias.”

5) Reference 1 and reference 23 are the same, please correct

Thank you for this comment- corrected

6) consider citing the following paper that focused on the outcomes of LT for HCC according to Metroticket score and Li-RADS 10.1111/tri.13983

Cited, as advised.

Round 2

Reviewer 1 Report

The authors have done a good job for revision of this manuscript. The discussion have been elaborated a little more and a little more in-depth analysis of the data have been done trying to find correlations with the data studied by the authors. 

Reviewer 2 Report

The Authors properly addressed all the comments

The revised version is fully suitable for publication